# Self-supervised Post-processing Method to Enrich Pretrained Word Vectors

**Hwiyeol Jo**
NAVER, Search US
hwiyeolj@gmail.com

## Abstract

Retrofitting techniques, which inject external resources into word representations, have compensated for the weakness of distributed representations in semantic and relational knowledge between words. However, the previous methods require additional external resources and strongly depend on the lexicon. To address the issues, we propose a simple extension of extrofitting, self-supervised extrofitting: extrofitting by its own word vector distribution. Our methods improve the vanilla embeddings on all of word similarity tasks without any external resources. Moreover, the method is also effective in various languages, which implies that our method will be useful in lexicon-scarce languages. As downstream tasks, we show its benefits in dialogue state tracking and text classification tasks, reporting better and generalized results compared to other word vector specialization methods.[1]

## 1 Introduction

Static word vectors are still widely used in natural language tasks despite the recent trends of contextualized models. For example, in a wide study of Dialogue State Tracking (DST) (Feng et al., 2021), contextualized models performed worse than static word embeddings. It seems reasonable that the contextualized models do not perform well in such a lack of context, which implies that static embeddings are still useful.

To make better word vectors, we focus on retrofitting ideas (also called word vector post-processing), which injects the semantic information from external resources by modifying the values of pretrained word vectors (Faruqui et al. (2015); Mrkšić et al. (2016a); *inter alia*). The benefits of post-processing methods are that (1) the methods can reflect additional resources into the word vectors without re-training, (2) the methods can be applied to any kinds of pretrained word vectors, and (3) retrofitting can make word vectors specialize in a specific task.

The previous studies focusing on explicit retrofitting have used manually defined or learned functions to make synonyms close and antonyms distant (see the details in §4). As a result, their approaches strongly rely on external resources.

Furthermore, we agree that making synonyms close together is somewhat reasonable, even though the synonyms have different nuances in some contexts. However, other kinds of relations, such as antonyms, should be further investigated. For instance, love and hate are generally grouped as antonyms. Most of the previous studies have made the words distant from each other, but the words definitely share the meaning of emotion in their representations. We thus conjecture that the methods are not generalized well.

In this paper, we propose word vector enrichment based on extrofitting (Jo and Choi, 2018):

- **Self-supervised extrofitting** that extends extrofitting for enriching word vectors without using external semantic lexicons. This method can resolve a limitation of post-processing approaches, which requires well-defined semantic lexicons. We highlight its usefulness in (relatively) lexicon-scarce languages.
- We report the effects of word vector post-processing on several downstream tasks to show the generalization of the word vectors. Our methods consistently improve model performances in the fundamental tasks. In contrast, other post-processing methods degrade the performance.

## 2 Preliminary

**Extrofitting** Extrofitting (Jo and Choi, 2018) expands word embedding matrix $W$ by concatenating

---

[1] http://github.com/hwiyeoljo/SelfExtro

$W$ with $r_w$:

$$\text{Expand}(W, c)$$
$$= W \oplus r_w \begin{cases} \text{mean}_{w \in c}(\mu_w) & \text{if } w \in \text{L} \\ \mu_w & \text{otherwise} \end{cases}$$

where $\mu_w$ is the mean value of elements in word vector $w$. L denotes semantic lexicons, and $c$ denotes the same class (synonym pairs). In other words, Expand makes an additional dimension per the word vector, and fill it with the same value if the pairs are synonym.

Next, Trans calculates transform matrix given the matrix $W$:

$$\text{Trans}(W, c)$$
$$= \text{argmax}_U \frac{|U^T \sum_c (\mu_c - \mu)(\mu_c - \mu)^T U|}{|U^T \sum_c \sum_i (x_i - \mu_c)(x_i - \mu_c)^T U|}$$

where $x$ is a word vector, $c$ is a class. The overall average of $x$ is $\mu$, and the class average in class $i$ is denoted by $\mu_i$. This formula finds a transform matrix $U$, which minimizes the variance within the same class and maximizes the variance between different classes. Each class is defined as the index of synonym pairs.

In the end, Extrofitting is formulated as follows:

$$\text{Extro}(W, c)$$
$$= \text{Trans}(\text{Expand}(W, c), c)^T \text{Expand}(W, c)$$

**Latent Semantic Analysis (LSA)** LSA (Landauer and Dumais, 1997) has been used to extract the relation of data through latent variables. LSA is based on Singular Value Decomposition (SVD), which decomposes a matrix as follows:

$$A = USV^T,$$

where $S$ is a diagonal matrix with singular values, and $U$ and $V$ are the orthogonal eigenvectors. We can select top-$k$ singular values to represent matrix $A$ in $k$-dimensional latent space. Then $U$ and $V$ are re-defined as $U_k \in \mathcal{R}^{N \times k}$ and $V_k^T \in \mathcal{R}^{k \times N}$, respectively, with diagonal matrix $S_k \in \mathcal{R}^{k \times k}$. When we use LSA for topic modeling, $A$ is defined as a term-document matrix. Then, $U_k S_k$ and $S_k V_k^T$ are considered as term vectors and document vectors in the k-dimensional latent space, respectively.

## 3 Self-supervised Extrofitting

We consider the word embedding matrix as the term-document matrix; The x-axis of the matrix is vocabulary and the y-axis is (unknown) semantic dimension. Thus, as researchers have use the decomposition of the term-document matrix to get the term-vectors in LSA, the decomposition of the term-semantic matrix can be considered to represent term-vectors ($US$) and semantic-vectors ($SV^T$). This intuition corresponds to the traditional methods in vector embedding, verified in Levy and Goldberg (2014); Allen et al. (2019).

With the idea, we first decompose word embeddings matrix $W$ to make latent representations as follows:

$$W_k = U_k S_k V_k^T$$

As described above, we can get term-vectors, which are word representations in $k$-dimensional latent space, by computing $U_k S_k$. Adjusting the dimension of latent space ($k$ in §2), we calculate semantically related words using cosine similarity. For every word, we calculate its cosine similarity to all other words in the vocabulary. If the similarity exceeds a predetermined threshold, we group the words. Note that a group can contain only a single word. This process is repeated iteratively for each ungrouped word until all words are clustered.

We use the set of semantically related words (c') as the class (synonym pairs, $c$) of extrofitting instead of semantic lexicons:

$$\text{SelfExtro}(W, c')$$
$$= \text{Trans}(\text{Expand}(W, c'), c')^T \text{Expand}(W, c')$$

To sum up, we use the principle of vanilla extrofitting (Jo and Choi, 2018), which utilizes LDA algorithms to group in-domain instances (in this case, word vectors). Simultaneously, the algorithm pushes out-domain instances apart. Instead of using pre-defined lexicons for extrofitting, we use the idea of LSA to get semantically related words. Even if the number of extracted word pairs are small or the words are not meaningful, the process of *making out-domain instances far* can make better representations.

In the experiments, we start with pretrained GloVe (Pennington et al., 2014) (if we do not mention it explicitly) and set a threshold that determines whether a pair of words are semantically related. We use a high threshold (0.9 of cosine similarity) since type II error is rather better than type I error.

## 4 Related Works

The first successful post-processing approach was **Retrofitting** (Faruqui et al., 2015), which modi-

| | GloVe (glove.42B) | | | | | | | fastText (wiki-news) | | | | | | |
|---|---|---|---|---|---|---|---|---|---|---|---|---|---|---|
| | W(s) | W(r) | RW | ME | SE | SL | SV | W(s) | W(r) | RW | ME | SE | SL | SV |
| Raw | .70 | .57 | .39 | .74 | .57 | .37 | .22 | .82 | .62 | .51 | .80 | .64 | .44 | .35 |
| SelfExtro, $k$=100 | .80 | .73 | .50 | .84 | .66 | .51 | .39 | .81 | .70 | .56 | .82 | .66 | .50 | .41 |
| SelfExtro, $k$=200 | .79 | .72 | .49 | .84 | .66 | .51 | .39 | .81 | .70 | .57 | .82 | .65 | .51 | .42 |
| SelfExtro, $k$=300 | .78 | .69 | .48 | .84 | .64 | .50 | .37 | .81 | .71 | .57 | .83 | .64 | .51 | .42 |
| +WordNet(=Extro) | .80 | .74 | .49 | .83 | .66 | .49 | .36 | .78 | .67 | .51 | .80 | .63 | .50 | .40 |
| vecmap | .67 | .53 | .31 | .72 | .52 | .35 | .20 | .81 | .62 | .49 | .80 | .59 | .44 | .34 |

Table 1: Spearman's correlation of self-supervised extrofitted pretrained word embeddings. $k$ denotes the dimension of latent space that is used to extract semantically related words (see §3). Ablation studies in threshold are presented in Appendix A.4.

fied word vectors by weighted averaging the word vectors with semantic lexicons. Extending from the simple idea, **Counter-fitting** (Mrkšić et al., 2016a) used both synonym pairs to collect word vectors and antonym pairs to make word vectors the gene from one another. Next, **Paragram embeddings** (Wieting et al., 2015) used synonyms and negative sampling to collect the word vectors. Borrowing attract-terms from the Paragram embeddings and adding repel-terms, **Attract-Repel** (Mrkšić et al., 2017) injected linguistic constraints into word vectors through predefined cost function with mono-/cross-lingual linguistic constraints. **Explicit Retrofitting (ER-CNT)** (Glavaš and Vulić, 2018) directly learned mapping functions of linguistic constraints with deep neural network architectures. They then used the functions to retrofit the word vectors. **Post-Specialization** (Vulić et al., 2018; Ponti et al., 2018) resolved the problem of the previous models that only updates words in external lexicons. Some works have used cross-lingual resources to get further semantic information (Vulić et al., 2019; Kamath et al., 2019).

While the previous methods utilized text-level resources, **vecmap** (Artetxe et al., 2018) used other word embeddings as external resources.[2] For fair comparisons, we input the same word vectors for source and target embeddings of the method.

Recently, BERT-based model LexFit (Vulić et al., 2021) was derived, but the method requires external resources.

## 5 Experiment 1: Word Similarity Tasks

**Settings.** Word similarity datasets consist of word pairs with human-rated similarity scores between the words and models calculate Spearman's correlation (Daniel, 1990) between the similarity

scores and the cosine similarity of the word vector pairs.

We use 6 datasets in English: WordSim353 (W(s) for similarity and W(r) for relation) (Finkelstein et al., 2001), RareWord (RW) (Luong et al., 2013), MEN-3k (ME) (Bruni et al., 2014), SemEval (SE) (Camacho-Collados et al., 2017), SimLex-999 (SL) (Hill et al., 2015), and SimVerb-3500 (SV) (Gerz et al., 2016).

**Results.** In Table 1, self-supervised extrofitting (SelfExtro) improves the performance on all the word similarity datasets when compared with the popular pretrained word vectors, GloVe (Pennington et al., 2014) and fastText (Bojanowski et al., 2016)[3]. The result implies the pretrained word vectors can be enriched by our method, which means it does not require any semantic lexicons to make better embeddings. In addition, SelfExtro shows better performances with the original extrofitting (+WordNet) on most of the evaluations.

We also investigate the extracted semantic information (see Appendix A.2). The extracted information hardly seems synonyms, but we can see that some similar words are grouped. As extrofitting affects all the word vectors, every word can benefit from the other words being enriched. In other words, although the number of extracted information is small, this simple extension makes extrofitting fully utilize its advantages.

Qualitative examples of self-supervised extrofitting are presented in Table 2. Although the representations lose similarity scores, the similar words become diverse and reasonable. Additional qualitative examples are shown in Appendix A.5.

Our proposed SelfExtro is also potentially useful when semantic lexicon resources are scarce, such as in many non-English languages. In Ta-

---

[2]We selected unsupervised version of vecmap because it performs better than identical version.

[3]word2vec could also be considered as a base model, but it takes too much time due to its large vocabulary size.

| Cue Word | Method | Top-10 Nearest Words(Cosine Similarity Score) |
|---|---|---|
| love | Vanilla | loved(.7745), i(.7338), loves(.7311), know(.7286), loving(.7263), really(.7196), always(.7193), want(.7192), hope(.7127), think(.7110) |
| | +SelfExtro | loved(.7152), loving(.6734), loves(.6489), **adore**(.6348), **passion**(.6333), **luv**(.6326), hope(.6256), i(.6250), want(.6209), **hate**(.6181) |
| hate | Vanilla | dont(.7318), stupid(.7193), hates(.7190), think(.7063), why(.6943), love(.6928), hating(.6927), hated(.6861), shit(.6847), know(.6825) |
| | +SelfExtro | hating(.6707), **hatred**(.6650), dont(.6580), hates(.6529), **dislike**(.6447), **despise**(.6309), hated(.6306), stupid(.6266), think(.6249), love(.6181) |
| forever | Vanilla | alive(.6478), gone(.6450), love(.6381), never(.6267), again(.6249), yours(.6238), life(.6171), alone(.6153), anymore(.6129), always(.6093) |
| | +SelfExtro | **eternally**(.6297), alive(.5814), yours(.5717), life(.5532), anymore(.5524), **eternity**(.5488), **eternal**(.5474), gone(.5447), **permanently**(.5446), again(.5414) |
| life | Vanilla | lives(.8053), living(.7134), things(.6869), way(.6852), mind(.6844), what(.6779), much(.6723), love(.6719), because(.6716), work(.6706) |
| | +SelfExtro | lives(.7801), living(.6538), **journey**(.6048), **everyday**(.6038), mind(.5922), things(.5920), **lifetime**(.5892), **happiness**(.5749), love(.5743), way(.5728) |

Table 2: List of top-10 nearest words of cue words in different post-processing methods. We report cosine similarity scores of words–`love`, `hate`, `forever`, and `life`. Underline and bold indicate the difference between the list from vanilla vectors and self-supervised extrofitted vectors, respectively.

| | CZ | IT(s/r) | | RU(s/r) | | GE(s/r) | |
|---|---|---|---|---|---|---|---|
| fastText | .240 | .377 | .188 | .428 | .275 | .335 | .169 |
| $k$=100 | .450 | .453 | .370 | .525 | .411 | .425 | .261 |
| $k$=200 | .443 | .444 | .359 | .545 | .426 | .431 | .261 |
| $k$=300 | .451 | .456 | .359 | .539 | .426 | .449 | .271 |

Table 3: Spearman's correlation of self-supervised extrofitting on WordSim datasets in 4 languages: Czech (CZ), Italian (IT), Russian (RU), and German (GE). $k$ denotes extracted semantic information from fastText in $k$-dimensional latent space.

| | CY | ET | FI | FR | HE | PL | RU | ES |
|---|---|---|---|---|---|---|---|---|
| fT | .187 | .213 | .455 | .314 | .371 | .283 | .305 | .385 |
| $k$=100 | .201 | .223 | .475 | .340 | .400 | .306 | .334 | .425 |
| $k$=200 | .209 | .215 | .463 | .339 | .399 | .299 | .306 | .423 |
| $k$=300 | .214 | .219 | .464 | .338 | .402 | .305 | .308 | .421 |

Table 4: Spearman's correlation of self-supervised extrofitting on Multi-SimLex datasets in 8 languages: Welsh (CY), Estonian (ET), Finnish (FI), French (FR), Hebrew (HE), Polish (PL), Russian (RU), and Spanish (ES). Other languages cannot be found in fastText.

| Model(Resource) | JGA with WordNet | JGA with A-R Lexicon |
|---|---|---|
| GloVe | .798(.03) | |
| Retrofitting(Syn) | .792(.03) | .793(.01) |
| Paragram(Syn) | .670(.05) | .657(.03) |
| Extro(Syn) | .821(.03) | .820(.01) |
| Counter-fit(Syn+Ant) | .625(.01) | .630(.03) |
| Att-Repel(Syn+Ant) | .671(.04) | .675(.03) |
| vecmap(-) | .772(.01) | |
| SelfExtro(-) | **.825(.02)** | |

Table 5: Joint goal accuracy and its standard deviation in WOZ 2.0 datasets. Despite the fact that we used the original Github code and data, the result is different from their report. Furthermore, our settings show different trends from what they reported.

ble 3 and Table 4, we use pretrained fastText for WordSim datasets in 4 languages and Multi-SimLex (Vulić et al., 2020) in 8 languages, respectively. `SelfExtro` significantly increases the performance in all the languages.

## 6 Experiment 2: Dialogue State Tracking

**Settings.** Previous works (Mrkšić et al., 2016a, 2017; Vulić et al., 2019) showed that word vector post-processing is useful for dialogue state tracking (DST) (Young et al., 2010). Using Neural Belief Tracker (NBT) (Mrkšić et al., 2016b), they have claimed that the performance on the task is related to word vector specialization in that the model has to identify the flow of dialogue with only a few words, which seems reasonable. Thus, we use the NBT[4] and check the model performances with our post-processed embeddings. Refer to the papers cited above for the details of DST.

**Results.** Table 5 shows the performance of DST in Wizard-of-Oz 2.0 dataset (Wen et al., 2017). The results show that the semantic specialization (e.g., Attract-Repel) does not increase the performance. In contrast, `Extro` and `SelfExtro` show better performance than vanilla GloVe.

We additionally experiment with the lexicons (both synonyms and antonyms) included in the Github of Attract-Repel[5]. It shows only a little

---

[4] https://github.com/nmrksic/neural-belief-tracker
[5] https://github.com/nmrksic/attract-repel

| Freezed Vectors | DBpedia | Yahoo(Up) | Yahoo(Low) | Yelp | AGNews | IMDB |
|---|---|---|---|---|---|---|
| GloVe | 98.31±0.12 | 71.97±0.76 | 49.35±0.51 | 61.55±0.35 | 90.66±0.79 | 87.47±1.93 |
| Retrofit(Syn) | 81.92±5.11 | 44.72±2.77 | 22.31±0.63 | 48.87±0.40 | 82.04±1.49 | 63.40±3.53 |
| Paragram(Syn) | 86.88±0.36 | 62.37±0.36 | 40.25±0.19 | 50.04±0.20 | 80.16±0.14 | 74.77±0.43 |
| Extro(Syn) | **98.52±0.05** | **72.94±0.10** | **50.01±0.39** | **62.70±0.06** | **91.36±0.29** | **89.40±0.20** |
| Counter-fit(Syn+Ant) | 73.26±1.29 | 59.70±0.19 | 37.80±0.08 | 47.97±0.45 | 76.47±1.45 | 66.54±0.48 |
| Att-Repel(Syn+Ant) | 87.52±0.12 | 62.51±0.41 | 39.75±1.15 | 50.08±0.29 | 79.80±0.93 | 74.71±0.37 |
| PostSpec(Syn+Ant) | 76.78±0.79 | 59.45±0.15 | 37.16±0.35 | 49.16±0.64 | 76.06±0.62 | 66.11±0.79 |
| vecmap(-) | 98.01±0.27 | 68.99±0.68 | 44.57±1.14 | 62.22±0.25 | 89.87±0.88 | 83.80±1.33 |
| SelfExtro(-) | **98.44±0.05** | **72.41±0.26** | **49.79±0.37** | **62.97±0.18** | **90.93±0.11** | **89.36±0.41** |
| *Trainable Vectors* | **DBpedia** | **Yahoo(Up)** | **Yahoo(Low)** | **Yelp** | **AGNews** | **IMDB** |
| GloVe | **98.61±0.05** | 73.45±0.70 | 49.30±0.61 | 63.01±0.42 | 91.54±0.45 | 88.82±0.55 |
| Retrofit(Syn) | 98.16±0.05 | 67.15±1.73 | 43.14±1.31 | 61.74±0.33 | 89.21±1.61 | 82.77±1.27 |
| Paragram(Syn) | 98.14±0.04 | 66.32±0.27 | 42.85±0.43 | 61.96±0.22 | 89.91±0.19 | 83.14±1.15 |
| Extro(Syn) | **98.61±0.06** | **73.50±0.19** | **50.43±0.30** | 63.22±0.19 | **91.78±0.29** | 89.74±0.28 |
| Counter-fit(Syn+Ant) | 98.07±0.10 | 63.74±0.53 | 40.83±0.20 | 61.89±0.17 | 90.09±1.23 | 83.78±1.84 |
| Att-Repel(Syn+Ant) | 98.11±0.07 | 66.05±0.42 | 42.68±0.43 | 61.80±0.21 | 90.05±0.18 | 83.74±1.30 |
| PostSpec(Syn+Ant) | 98.10±0.05 | 63.88±0.36 | 40.49±0.51 | 61.84±0.13 | 90.48±0.38 | 84.54±0.38 |
| vecmap(-) | 98.01±0.27 | 68.99±0.68 | 44.57±1.14 | 62.22±0.25 | 89.87±0.88 | 83.80±1.33 |
| SelfExtro(-) | **98.58±0.02** | **73.02±0.15** | **50.05±0.23** | **63.25±0.25** | **91.61±0.08** | **89.98±0.26** |

Table 6: 10 times average accuracy initialized with the post-processed word vectors. We experiment with our methods in 2 different settings: fixed word vectors and trainable word vectors.

performance gain; we thus guess that the difference in DST performance comes from lexical resources or fine-tuning of NBT rather than specialization.

## 7 Experiment 3: Text Classification

**Settings.** We experiment with our methods in 2 different settings: fixed word vectors and trainable word vectors. When the word vectors are fixed, we can evaluate the usefulness of the word vectors per se. When the word vectors are trainable, we can see the improvement of the model performance in a conventional training setting. The dataset and classifier are described in Appendix A.3.

**Results.** We report the performances in Table 6. The classifier initialized with `SelfExtro` outperforms the vanilla GloVe and performs on par with the original extrofitting. On the other hand, although the previous word vector post-processing methods specialize well on a domain-specific task, the approaches failed to be generalized; they show large performance deterioration despite the added *general* semantic information.

## 8 Discussion

The specialized word vectors do not warrant better performance. We conjecture that the methods trade off catastrophic forgetting against semantic specialization even though the previous methods successfully specialize their vectors into given lexicons, losing the pretrained information encoded in

the original GloVe. It affects the performance of the fundamental tasks, which are largely degraded. On the other hand, our method enriches the word vector in general, resulting in marginal but certain improvements.

Contextual representations provide substantial improvements in NLP tasks, but the representations lack semantic information due to their nature. Since the context and the semantics are different types of information that can enrich representations, we believe our approaches might further improve the contextual representations.

## 9 Conclusion

We develop a self-supervised retrofitting model that enriches word vectors without semantic lexicons. The method utilizes its own distribution of word vectors to get better representations. In the Exp. 1, we show that our method can improve the performance on word similarity tasks and present qualitative examples. It can be applied to other pretrained word vectors, resulting in better performances on all the word similarity datasets. `SelfExtro` also has potential advantages in lexicon-scarce languages. In the Exp. 2 and 3, we presented the effect of post-processing methods on downstream tasks. Our method shows marginal but certain improvements, while other post-processed word vectors largely degrade the performances, which seems the result of losing generalization.

## 10 Limitations

**Usefulness compared with contextual embeddings** Contextual embeddings are recently dominant in NLP tasks, whereas static word embeddings have become less frequently used. However, it does not mean static word embeddings are not useful. Although we can assume that the amount of pre-training dataset and training resources are similar, the cost of inference is much cheaper at static word embeddings. Furthermore, static word embeddings perform better when a task lacks enough context (e.g., word similarity tasks). It will be interesting future work to retrofit contextual embeddings, but it is out of scope in this paper.

**The use of antonyms** Although we believe that there are no definitely opposite meanings of the word (e.g., love and hate share the sense of emotion), previous works (see §4) that utilize antonyms showed significant improvement in word similarity tasks. However, compared to the methods, self-supervised extrofitting *explicitly* considers synonyms only, but *implicitly* expects antonyms to be distant while maximizing the variances of in-class/out-of-class word vectors. Also, the process of self-supervised extrofitting makes it hard to incorporate other kinds of word relations.

**The method is only a simple linear projection** Both extrofitting and self-supervised extrofitting use *linear* projections in enriching vectors, following a traditional NLP method. The linear model might not be the best to reflect word relations in vector spaces, but we believe that it is a simple yet effective method, as we still calculate lots of things (e.g., distance) in a linear way.

## Acknowledgement

The author would like to thank all the reviewers in several rounds of submission, a total of 5 years. Lastly, I am grateful to Alice Lee for her help in qualitative analysis.

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

## A Appendix

### A.1 Data Resources

#### A.1.1 Pretrained word vectors

Pretrained word vectors include words composed of n-dimensional float vectors. One of the major pretrained word vectors we used is **GloVe** (Pennington et al., 2014) glove.42B.300d. Even though many word embedding algorithms and pretrained word vectors have been suggested, GloVe is still being used as a strong baseline on word similarity tasks (Cer et al., 2017; Camacho-Collados et al., 2017). We also use **fastText** (Bojanowski et al., 2016), and **Paragram** (Wieting et al., 2015) as resources and baseline of self-supervised extrofitting.

#### A.1.2 Semantic Lexicon

As an external semantic lexicon, we use **Word-Net** (Miller, 1995), which consists of approximately 150,000 words and 115,000 synsets pairs. We use Faruqui et al.'s WordNet$_{all}$ lexicon, comprised of synonyms, hypernyms, and hyponyms. Some recent works built their own or used large lexicons like BabelNet (Navigli and Ponzetto, 2012), but in order to observe the effect of post-processing algorithm rather than the power of lexicons, we use the (relatively) small lexicon.

For fair comparisons, we replace the previous works' lexicon with WordNet$_{all}$ lexicon. If the models require antonyms, we use the antonyms pairs, which are uploaded in their Github.

### A.2 Example of Extracted Relation

| Threshold | Grouped Words |
|---|---|
| ≥ .90 | points225 points203 points166 points253 ⋯ |
| | 17:20:18 09:22:30 10:22:49 07:33:04 ⋯ |
| | review10/15/2012 review3/31/2013 |
| | review11/30/2012 review12/18/2012 ⋯ |
| ≥ .99 | !!! !! |
| | january february |
| | range emailicon linkicon |

Table 7: Examples of extracted related word groups. This results are extracted from 100 dimensional latent space in GloVe.

### A.3 Further Details in Experiment 3

**Datasets.** We use 5 classification datasets; DB-pedia ontology (Lehmann et al., 2015), Yahoo!Answers (Chang et al., 2008)[6], YelpRe-

views (Zhang et al., 2015), AGNews, and IMDB (Maas et al., 2011). We utilize Yahoo!Answer dataset for 2 different tasks, which are classifying upper-level categories and classifying lower-level categories, respectively. We use all the words tokenized by space as inputs. The data information is described in Table 8.

**Classifier.** We build TextCNN (Kim, 2014) rather than use a classifier based on Bag-of-Words (BoW), as Faruqui et al. did, in order to process word sequences. The classifier consists of 2 convolutional layers with the channel size of 32 and 16, respectively. We adopt the multi-channel approach, implementing 4 different sizes of kernels–2, 3, 4, and 5. We concatenate them after every max-pooling layer. The learned kernels go through an activation function, ReLU (Hahnloser et al., 2000), and max-pooling. We set the dimension of word embedding to 300, optimizer to Adam (Kingma and Ba, 2014) with learning rate 0.001, and use early-stopping if validation accuracy does not increase over 5 epochs.

### A.4 Ablation Study in Threshold

When the latent dimension is 300, following the best performance:

### A.5 Qualitative Examples

The additional qualitative examples are presented in Table 12.

---

[6]`https://cogcomp.seas.upenn.edu/page/resource_view/89` Note that Chang et al. (2008) said the dataset has 20 top-level categories but actually it has 3

duplicated top-level categories because of errors.

|  | DBpedia | Yahoo(Up) | Yahoo(Low) | AGNews | Yelp | IMDB |
|---|---|---|---|---|---|---|
| #Train | 560,000 | 133,703 | 133,703 | 120,000 | 650,000 | 25,000 |
| #Test | 70,000 | 23,595 | 23,595 | 7,600 | 50,000 | 25,000 |
| #Class | 14 | 17 | 280 | 4 | 5 | 2 |
| #Vocab | 626,717 | 154,142 | 154,142 | 66,049 | 198,625 | 47,113 |

Table 8: The data information used in text classification. YahooAnswer dataset is used for 2 different tasks, which are to classify upper-level categories and to classify lower-level categories, respectively. The vocabulary size can be slightly different due to the predefined special tokens such as `none` and `out-of-vocabulary`

| | Word Similarity Task | | | | | | |
|---|---|---|---|---|---|---|---|
| | W(s) | W(r) | RW | ME | SE | SL | SV |
| GloVe | .70 | .57 | .39 | .74 | .57 | .37 | .28 |
| threshold=.95 | .72 | .60 | .39 | .76 | .58 | .42 | .28 |
| threshold=.90 | .78 | .69 | .48 | .84 | .64 | .50 | .37 |
| threshold=.50 | .71 | .59 | .38 | .75 | .56 | .39 | .23 |

Table 9: Spearman's correlation of self-supervised extrofitted pretrained word embeddings according to threshold.

| Model(threshold) | DST |
|---|---|
| GloVe | .798 |
| SelfExtro(threshold=.95) | .800 |
| SelfExtro(threshold=.90) | .825 |
| SelfExtro(threshold=.50) | .811 |

Table 10: Joint goal accuracy according to threshold

| Freezed Vectors | DBpedia | Yahoo(Up) | Yahoo(Low) | Yelp | AGNews | IMDB |
|---|---|---|---|---|---|---|
| GloVe | 98.31 | 71.97 | 49.35 | 61.55 | 90.66 | 87.47 |
| SelfExtro(thr=.95) | 98.35 | 71.61 | 49.07 | 61.01 | 90.78 | 87.23 |
| SelfExtro(thr=.90) | 98.44 | 72.41 | 49.79 | 62.97 | 90.93 | 89.36 |
| SelfExtro(thr=.50) | 98.49 | 72.74 | 50.80 | 62.10 | 90.94 | 88.33 |
| Trainable Vectors | DBpedia | Yahoo(Up) | Yahoo(Low) | Yelp | AGNews | IMDB |
| GloVe | 98.61 | 73.45 | 49.30 | 63.01 | 91.54 | 88.82 |
| SelfExtro(thr=.95) | 98.42 | 72.08 | 49.17 | 61.83 | 91.03 | 86.80 |
| SelfExtro(thr=.90) | 98.58 | 73.02 | 50.05 | 63.25 | 91.61 | 89.98 |
| SelfExtro(thr=.50) | 98.58 | 73.35 | 50.66 | 62.64 | 91.67 | 88.64 |

Table 11: Text classification accuracy according to threshold

| Cue Word | Method | Top-10 Nearest Words(Cosine Similarity Score) |
|---|---|---|
| soo | Vanilla | sooo(.8394), soooo(.7938), sooooo(.7715), soooooo(.7359), sooooooo(.6844), haha(.6574), hahah(.6320), damn(.6247), omg(.6244), hahaha(.6219) |
| | +SelfExtro | sooo(.8196), soooo(.7743), sooooo(.7576), soooooo(.7304), sooooooo(.6852), **sooooooooo(.6342), soooooooooo(.6314), sooooooooooo(.6003)** **tooo(.5967), soooooooooooo(.5869)** |
| elaborate | Vanilla | intricate(.7244), elaborately(.6238), extravagant(.6223), complicated(.6089), lavish(.5714), formal(.5643), sophisticated(.5639), detailed(.5623), ornate(.5619), simple(.5566) |
| | +SelfExtro | intricate(.7002), elaborately(.6329), extravagant(.6109), complicated(.5651), ornate(.5586), lavish(.5533), **grandiose(.5452)**, formal(.5326), detailed(.5235), **fanciful(.5225)** |
| gratitude | Vanilla | thankfulness(.7113), generosity(.6967), kindness(.6917), appreciation(.6860) compassion(.6695), admiration(.6672), grateful(.6420), thankful(.6385), heartfelt(.6368), sympathy(.6211) |
| | +SelfExtro | thankfulness(.7648), generosity(.7001), kindness(.6975), appreciation(.6934), admiration(.6830), compassion(.6738), **gratefulness(.6444)**, **reverence(.6361)**, thankful(.6313), sympathy(.6304) |
| jubilate | Vanilla | exsultate(.5262), deum(.4763), exultate(.4545), motet(.4054), excelsis(.3777), deo(.3721), cantata(.3651), alleluia(.3625), stabat(.3558), laudamus(.3485) |
| | +SelfExtro | exsultate(.5111), exultate(.4449), deum(.4269), motet(.4093), deo(.3703), **choir(.3670)**, excelsis(.3626), **mozart(.3594)**, stabat(.3591), cantata(.3532) |
| elated | Vanilla | overjoyed(.7712), ecstatic(.7045), thrilled(.6576), exhilarated(.6533), gratified(.6431), enthused(.6214), saddened(.6075), flabbergasted(.6060), disheartened(.5989), giddy(.5984) |
| | +SelfExtro | overjoyed(.8018), ecstatic(.7424), thrilled(.7320), **excited(.6894)**, exhilarated(.6758), **delighted(.6733)**, gratified(.6690), **relieved(.6569)**, enthused(.6564), **dismayed(.6342)** |
| mono-saccharide | Vanilla | disaccharide(.6470), monosaccharides(.5152), oligosaccharide(.4911), galactose(.4686), 5-carbon(.4609), n-acetylglucosamine(.4282), saccharide(.4223), sucrose(.4215), mannose(.4196), disaccharides(.4107) |
| | +SelfExtro | disaccharide(.6329), monosaccharides(.4888), oligosaccharide(.4844), 5-carbon(.4617), galactose(.4414), n-acetylglucosamine(.4227), sucrose(.4206), disaccharides(.4050), **carbohydrate(.4045)**, **moieties(.4033)** |
| outlook | Vanilla | excel(.5686), recovery(.5457), export(.5368), forecast(.5229), forecasts(.5060), contacts(.5025), powerpoint(.5025), exchange(.5011), microsoft(.4944), import(.4816) |
| | +SelfExtro | excel(.5145), **mapssevere(.5054)**, icalendargoogle(.4955), export(.4883), contacts(.4877), recovery(.4753), **advisorieshourly(.4673)**, **mailbox(.4661)**, **vcard(.4612)**, **thunderbird(.4599)** |
| vain | Vanilla | foolish(.6229), futile(.6104), selfish(.5340), vainly(.5242), fruitless(.5133), thy(.4981), arrogant(.4976), useless(.4844), righteous(.4833), thou(.4830) |
| | +SelfExtro | futile(.6273), foolish(.6079), vainly(.5454), fruitless(.5338), selfish(.5143), useless(.5044), **pointless(.4910)**, **hopeless(.4846)**, **attempt(.4797)**, **presumptuous(.4780)** |
| prioritize | Vanilla | prioritise(.8064), prioritizing(.7426), priorities(.6323), prioritising(.5995), evaluate(.5950), strategize(.5933), proactively(.5862), analyze(.5800), assess(.5764), identify(.5633) |
| | +SelfExtro | prioritise(.8322), prioritizing(.7415), priorities(.6390), prioritising(.6225), proactively(.6075), strategize(.5990), evaluate(.5946), analyze(.5919), **prioritized(.5880)**, **manage(.5783)** |
| nomad | Vanilla | nomads(.4911), drifter(.4735), nomadic(.4574), vagabond(.4233), zen(.4161), muvo2(.4054), gypsy(.4054), jukebox(.3997), vulcan(.3906), adventurer(.3869) |
| | +SelfExtro | nomads(.5254), drifter(.4916), nomadic(.4841), muvo2(.4631), vagabond(.4422), **traveler(.4246)**, **traveller(.4216)**, zen(.4150), jukebox(.4110), **wanderer(.4073)** |
| junction | Vanilla | junctions(.5729), near(.4985), road(.4936), intersection(.4923), creek(.4911), highway(.4878), exit(.4853), bridge(.4811), rd(.4779), jct(.4772) |
| | +SelfExtro | junctions(.5822), jct(.5099), highway(.4787), **motorway(.4781)**, road(.4744), exit(.4738),**hwy(.4710)**, rd(.4667), intersection(.4628), **heafford(.4596)** |
| moss | Vanilla | peat(.5360), brady(.5090), reed(.4998), brown(.4963), fern(.4945), green(.4920), woods(.4872), pine(.4856), jones(.4840), campbell(.4824), |
| | +SelfExtro | **lichen(.5628)**, **sphagnum(.5463)**, **spagnum(.5457)**, peat(.5323), **lichens(.5264)**, **mosses(.5224)**, fern(.4934), **welker(.4852)**, pine(.4846), **ferns(.4805)**, |

Table 12: List of top-10 nearest words of cue words in different post-processing methods. We report cosine similarity scores of random words. Underline and bold indicate the difference between the list from vanilla vectors and self-supervised extrofitted vectors, respectively.