# OpenReview forum: "Self-supervised Post-processing Method to Enrich Pretrained Word Vectors"
_EMNLP/2023/Conference — EMNLP 2023 Findings_

### Official Review · Reviewer_dUyB · 2023-08-02

**Soundness:** 3

**Excitement:**

4: Strong: This paper deepens the understanding of some phenomenon or lowers the barriers to an existing research direction.

**Paper Topic And Main Contributions:**

This paper introduces a new method for self-supervised extrofitting that doesn't require an external semantic resource. This makes this method applicable to any pretrained embedding set for any language, even if that language does not have readily available semantic resources. The authors test their approach on word similarity, dialogue state tracking, and text classification tasks and show comparable (if not superior) performance to traditional extrofitting methods which do require an external semantic resource.

**Questions For The Authors:**

Did you perform any kind of significance testing in your experiments? If not, why?

Table 6 presents groups of words that share a minimum cosine similarity in the latent space. Is this similarity compared against some anchor word embedding (i.e. every vword in the group is at least the minimum similarity with the anchor word)? Or are all pairs within that group a minimum similarity with each other? If the former, what are the anchor words for each of those groups? If the latter, how did you find these groups?

The example groups given in Table 6 seem to largely consist of non-words (i.e. may of them appear to be the result of bad text sanitation in the original GloVe embeddings) and doesn't give confidence in the extracted word groups. Did you perform any qualitative analysis on the extracted groups?

**Reasons To Accept:**

- relatively simple way to improve performance of pretrained word embeddings
- language independent
- reduces need for external resources

**Reasons To Reject:**

The provided discussion of experimental results is very surface level and doesn't provide the reader with little insight into how the proposed method enhances performance and what the limitations or pitfalls of this approach might be.

**Reproducibility:**

2: Would be hard pressed to reproduce the results. The contribution depends on data that are simply not available outside the author's institution or consortium; not enough details are provided.

**Reviewer Confidence:**

2: Willing to defend my evaluation, but it is fairly likely that I missed some details, didn't understand some central points, or can't be sure about the novelty of the work.

**Typos Grammar Style And Presentation Improvements:**

S1P2 Line 035 - "any kinds" -> any kind

Some issues with column width in table 7

Forcing A4 into two columns would result in less wasted space

---

> ### Author Rebuttal · Authors · 2023-08-28
>
> Thanks for letting us know the concerns with too brief explanations of the results.
> We hope this rebuttal could convince you that this paper can be improved significantly until camera-ready with the help of your comments.
> Please kindly understand that this is mainly due to the page limit; we hope you could relieve the concerns through this response.
>
> [Weak1]The provided discussion of experimental results is very surface level
> - In the current version of this paper, we attempted to present empirical results as much as possible rather than investigate the word vectors with a closer look, which can appear to be cherry-picked
> - We will add more reasons/explanations, like qualitative analysis. Please see the appendix of this rebuttal below.
>
> [Question 1] significance testing in your experiments
> - The experiments (except for intrinsic evaluation) include standard deviations, and the results are statistically meaningful.
> - In the case of word similarity tasks, self-extrofitting generates consistent representations in the same setting.
>
> [Question 2] How to group the synonym words
> - "are all pairs within that group a minimum similarity with each other?" is right.
> - To be specific, the processes are:
>   - For every word (referred to as an anchor word), we calculate its cosine similarity to all other words in the vocabulary.
>   - If the similarity exceeds a predetermined threshold, we group the words.
>     - In this process, a group can contain only a single word.
>   - This process is repeated iteratively for each ungrouped word until all words are assigned to groups.
>     - It means the next anchor word will not assign to existing group; it always generates new group.
> - It's important to note that the order of anchor word selection doesn't affect the final groups due to the symmetric nature of cosine similarity.
>
> [Question 3] Quality of extracted word groups
> - When we observe the extracted word groups, most are similar to Table 6. (e.g., timestamp, the name of months, numbers etc.)
> - Although the extracted word groups may not seem linguistically meaningful at first glance, they serve as a crucial mechanism for the effect of self-extrofitting.
>   - By clustering seemingly unimportant words and scattering the other words together, we facilitate the formation of more meaningful connections between words, ultimately enhancing the vector representations.
> - Due to this advantage, we claim that the extracted word groups itself do not need to be meaningful.
> - If you are interested in the quality of the result, please refer to the Appendix of this rebuttal.
>
> [Appendix]
> - We do not want to say that this only one example of qualitative analysis means our method is superior, rather to say the result is meaningful, so please see it as a reference:
> - For example, even if we used the extracted word groups (which seems linguistically meaningless), the similar words to 'love' in the vector spaces are as follows:
>     - Vanilla glove (glove.42B.300d.txt):
>
>     [('loved', 0.7745193839073181), ('i', 0.733807384967804), ('loves', 0.7310654520988464), ('know', 0.7285589575767517), ('loving', 0.7262834310531616), ('really', 0.7195589542388916), ('always', 0.7193377017974854), ('want', 0.7191829681396484), ('hope', 0.712653636932373), ('think', 0.7109541893005371)]
>     - SelfExtro:
>
>     [('loved', 0.715188205242157), ('loving', 0.6733922362327576), ('loves', 0.6488849520683289), ('adore', 0.6347748637199402), ('passion', 0.6332944631576538), ('luv', 0.6325735449790955), ('hope', 0.6255802512168884), ('i', 0.6250261068344116), ('want', 0.6209121346473694), ('hate', 0.6181126832962036)]
>     - Although the representations lose similarity scores, the list of close words become diverse and reasonable; we want to claim it as better-generalized representation.

---

### Official Review · Reviewer_fe3o · 2023-08-04

**Typos Grammar Style And Presentation Improvements:** Line no 119
**Soundness:** 4

**Excitement:**

4: Strong: This paper deepens the understanding of some phenomenon or lowers the barriers to an existing research direction.

**Paper Topic And Main Contributions:**

- The paper proposes a new approach called self-retrofitting of word vectors without depending on external lexicons and also presented the results on languages where the resources are less.
- Instead of using the external semantic lexicons to find the related words, they use the latent semantic analysis method to find the related words and applying the standard retrofitting approach on top of it.
- The experiments are conducted across different tasks of word similarity, dialogue state tracking and text classification and obtained better results
- The paper is written well and the related work is comprehensive that compares and contrasts with this work
- The limitations of the work are also mentioned in detail


**Questions For The Authors:**

Question A: Are the results presented in the paper are statistically significant?
Question B: Are there any other metrics used for text classification other than the accuracy? If yes, how do the results change?

**Reasons To Accept:**

- The approach in the paper will be used for languages where the resources are less
- The approach doesn't depend on the presence of external semantic lexicons
- The results presented in the paper can be easily reproducible

**Reasons To Reject:**

- Even though the results are better or comparable to the existing approaches, it's not mentioned in the paper whether the results are statistically significant

**Reproducibility:**

5: Could easily reproduce the results.

**Reviewer Confidence:**

4: Quite sure. I tried to check the important points carefully. It's unlikely, though conceivable, that I missed something that should affect my ratings.

---

> ### Author Rebuttal · Authors · 2023-08-28
>
> Thanks for your interest in this paper and for letting us know its weaknesses. We hope this rebuttal answers the questions.
>
> [Weak1 & Question A] it's not mentioned in the paper whether the results are statistically significant
> - Since the results of self-extrofitting make consistent vector representations, we do not need to measure the significance test in intrinsic tasks.
> - However, the results of the other downstream tasks can be different because of weight initialization; we thus denote every standard deviation.
> - As you can see through the reports, most of the results are meaningful.
>
> [Question B] Are there any other metrics used for text classification other than the accuracy? If yes, how do the results change?
> - F1 metrics could be used for text classification, but the trends are the same with accuracy since the text classification datasets are balanced (except for YahooAnswers)
> - In the case of YahooAnswers, it also showed the same trends.

---

### Official Review · Reviewer_FCUs · 2023-08-05

**Soundness:** 3

**Excitement:**

3: Ambivalent: It has merits (e.g., it reports state-of-the-art results, the idea is nice), but there are key weaknesses (e.g., it describes incremental work), and it can significantly benefit from another round of revision. However, I won't object to accepting it if my co-reviewers champion it.

**Paper Topic And Main Contributions:**

This paper describes an approach to semantic specialisation of word vectors, i.e. an optimisation method that can be applied to off-the-shelf word vectors in order to improve their usefulness for downstream tasks. The approach described here ("self-supervised extrofitting") is based on the extrofitting method of Jo and Choi (2018) but differs from that method and most prior methods by not requiring a manually curated lexicon to group words by similarity. The authors show that despite not using a lexicon, the resulting word vectors have comparable or better quality to those derived with the extrofitting method.

**Reasons To Accept:**

There are multiple experiments to evaluate intrinsic and extrinsic quality, and the overall picture is that the proposed method has some benefits:
- It is a nice result that the self-supervised method can achieve similar quality to the lexicon-driven extrofitting method; as the authors observe this can be helpful in languages where there are limited lexical resources.
- While the authors tried and failed to reproduce previous results on the DST dataset (I appreciate their honesty in stating this), I note the SelfExtro results are better than those reported in the original Attract-Repel paper.

Practitioners who have a need for non-contextual word embeddings (e.g. for efficiency reasons) may find this approach a useful one.

**Reasons To Reject:**

- The authors don't convey any intuitions about why this method works well. Unlike other specialisation methods, the approach in this paper doesn't introduce any new information that's not already present in the vectors. For example, is it necessary to perform SVD-based dimensionality reduction on vectors that are already highly compressed?

- The intrinsic evaluation presents results that are far from the state of the art, e.g. the SimLex-999 homepage indicates that the best score in 2016 was 0.76, compared to 0.50 reported here. That doesn't mean the method is not useful, but there is some important context missing.

**Reproducibility:**

4: Could mostly reproduce the results, but there may be some variation because of sample variance or minor variations in their interpretation of the protocol or method.

**Reviewer Confidence:**

4: Quite sure. I tried to check the important points carefully. It's unlikely, though conceivable, that I missed something that should affect my ratings.

**Typos Grammar Style And Presentation Improvements:**

p. 2 typo "have use the decomposition"

Footnote 2: "it performs better than another version" - please be specific, which other version?

---

> ### Author Rebuttal · Authors · 2023-08-28
>
> We appreciate your critical comments (especially [Weak1]). It will help clarify the idea behind this work.
> We hope this rebuttal could convince you that this paper can be much improved until camera-ready with the help of your comments.
>
> [Weak1] The authors don't convey any intuitions about why this method works well
>
> - We agree that we needed to clarify the explanations in the main page. The explanations were scattered and briefly summarized in the Limitation Section.
> - We want to clarify that the intuition is:
>   - LDA algorithms group in-domain instances (in this case word vectors), together.
>   - Simultaneously, they push out-domain instances apart, which aligns with the principle of vanilla extrofitting.
>   - Therefore, even though we have only a small amount of lexicons to determine in/out-domain, the process of "making out-domain instances far", in particular, could result in better representations.
>   - To do this, we combine extrofitting with a simple SVD trick in order to get lexicons.
>
> [Weak2] The intrinsic evaluation presents results that are far from the state of the art
> - First of all, we appreciate your understanding that "That doesn't mean the method is not useful"
> - Since we do not have any training process and external resources to fit on SimLex, it seems natural that the performance of our method is behind the state-of-the-art. Furthermore, the goal of this method is to get better-generalized representations.
> - However, extending this method for vector specialization could be an interesting future work.
>
> [Minor1] Footnote 2: "it performs better than another version" - please be specific, which other version?
> - Thank you for the correction. vecmap has identical version and unsupervised version, so it refers identical version.

---

### Meta-Review · Area_Chair_W3sQ · 2023-09-19

**Recommendation:** 3

**Metareview:**

The paper introduces a word on semantic specialisation of word vectors, based on the extrofitting method, without requiring any external lexicon or manually curated resources. The proposed method improves its baseline on various benchmarks and languages, showing the quality of the approach.
After author rebuttal, the reviewers, in general, agree that the work is simple, language independent, and it improves the performance of vanilla word vectors without the need for an external lexicon. They appreciated the author response, however, the paper would need a better explanation of the intuition behind the proposed method, so the clarity of the paper should be improved, as it reads more like an incremental work.

---

### Decision · Program_Chairs · 2023-10-07

**Decision:**

Accept-Findings

**Comment:**

The paper introduces a word on semantic specialisation of word vectors, based on the extrofitting method, without requiring any external lexicon or manually curated resources. The proposed method improves its baseline on various benchmarks and languages, showing the quality of the approach.
After author rebuttal, the reviewers, in general, agree that the work is simple, language independent, and it improves the performance of vanilla word vectors without the need for an external lexicon. They appreciated the author response, however, the paper would need a better explanation of the intuition behind the proposed method, so the clarity of the paper should be improved, as it reads more like an incremental work.